

# High-resolution inversion of OMI formaldehyde columns to quantify isoprene emission on ecosystem-relevant scales: application to the Southeast US

Jennifer Kaiser[1], Daniel J. Jacob[1,2], Lei Zhu[1], Katherine R. Travis[1,*], Jenny A. Fisher[3,4], Gonzalo González Abad[5], Lin Zhang[6], Xuesong Zhang[7], Alan Fried[8], John D. Crounse[9], Jason M. St. Clair[9,**], and Armin Wisthaler[10,11]

[1]John A. Paulson School of Engineering and Applied Sciences, Harvard University, Cambridge, MA, USA
[2]Department of Earth and Planetary Sciences, Harvard University, Cambridge, MA, USA
[3]Centre for Atmospheric Chemistry, School of Chemistry, University of Wollongong, Wollongong, NSW, Australia
[4]School of Earth and Environmental Sciences, University of Wollongong, Wollongong, NSW, Australia
[5]Harvard–Smithsonian Center for Astrophysics, Cambridge, MA, USA
[6]Laboratory for Climate and Ocean-Atmosphere Studies, Department of Atmospheric and Oceanic Sciences, School of Physics, Peking University, Beijing 100871, People's Republic of China
[7]Department of Physics, University of Toronto, Toronto, Ontario, Canada
[8]Institute for Arctic and Alpine Research, University of Colorado, Boulder, CO, USA
[9]Division of Geological and Planetary Sciences, California Institute of Technology, Pasadena, CA, USA
[10]Institute for Ion Physics and Applied Physics, University of Innsbruck, Innsbruck, Austria
[11]Department of Chemistry, University of Oslo, Oslo, Norway
[*]now at Department of Civil and Environmental Engineering, Massachusetts Institute of Technology, Cambridge, MA, USA.
[**]now at Atmospheric Chemistry and Dynamics Laboratory, NASA Goddard Space Flight Center, Greenbelt, MD, USA & Joint Center for Earth Systems Technology, University of Maryland Baltimore County, Baltimore, MD, USA

Correspondence to: Jennifer Kaiser (jkaiser@seas.harvard.edu)

**Abstract.** Isoprene emissions from vegetation have a large effect on atmospheric chemistry and air quality. 'Bottom-up' isoprene emission inventories used in atmospheric models are based on limited vegetation information and uncertain land cover data, leading to potentially large errors. Satellite observations of atmospheric formaldehyde (HCHO), a high-yield isoprene oxidation product, provide 'top-down' information to evaluate isoprene emission inventories through inverse analyses. Past inverse analyses have however been hampered by uncertainty in the HCHO satellite data, uncertainty in the time- and $NO_x$-dependent yield of HCHO from isoprene oxidation, and coarse resolution of the atmospheric models used for the inversion. Here we demonstrate the ability to use HCHO satellite data from OMI in a high-resolution inversion to constrain isoprene emissions on ecosystem-relevant scales. The inversion uses the adjoint of the GEOS-Chem chemical transport model at $0.25° \times 0.3125°$ horizontal resolution to interpret observations over the Southeast US in August-September 2013. It takes advantage of concurrent NASA SEAC[4]RS aircraft observations of isoprene and its oxidation products including HCHO to validate the OMI HCHO data over the region, test the GEOS-Chem isoprene oxidation mechanism and $NO_x$ environment, and independently evaluate the inversion. This evaluation shows in particular that local model errors in $NO_x$ concentrations propagate to biases in inferring isoprene emissions from HCHO data. It is thus essential





to correct model $NO_x$ biases, which was done here using SEAC$^4$RS observations but can be done more generally using satellite $NO_2$ data concurrently with HCHO. We find in our inversion that isoprene emissions from the widely-used MEGAN v2.1 inventory are biased high over the Southeast US by 40% on average, although the broad-scale distributions are correct including maximum emissions in Arkansas/Louisiana and high base emission factors in the oak-covered Ozarks of Southeast

Missouri. A particularly large discrepancy is in the Edwards Plateau of Central Texas where MEGAN v2.1 is too high by a factor of 3, possibly reflecting errors in land cover. The lower isoprene emissions inferred from our inversion, when implemented into GEOS-Chem, decrease surface ozone over the Southeast US by 1–3 ppb and decrease the isoprene contribution to organic aerosol from 40% to 20%.

# 1 Introduction

Isoprene from vegetation comprises about one third of the global emission of volatile organic compounds (VOCs), and emissions in the southeastern United States during summertime are some of the highest in the world (Guenther et al., 2006). Isoprene oxidation fuels tropospheric ozone formation in both rural and urban regions (Monks et al., 2015), and isoprene oxidation products contribute significantly to organic aerosol (Carlton et al., 2009). Regional air-quality predictions are heavily dependent on isoprene emission estimates (Pierce et al., 1998; Fiore et al., 2005; Hogrefe et al., 2011; Mao et al.,

2013). The uncertainty in isoprene emissions on a global scale is estimated to be factor of 2 or more, with larger uncertainties on local-to-regional scales (Guenther et al., 2012). Here, we use observations of formaldehyde (HCHO) columns from the satellite-based Ozone Monitoring Instrument (OMI) in the first high-resolution adjoint-based inverse analysis of isoprene emissions at ecosystem-relevant scales, taking advantage of detailed chemical measurements available over the Southeast US to demonstrate the capability of the satellite-based inversion.

Process-based "bottom-up" isoprene emission inventories are constructed by estimating base leaf-level emission rates for individual plant functional types (PFTs), mapping them onto gridded PFT distributions, and applying factor dependences on environmental variables (temperature, insolation, leaf area index and leaf age, soil moisture) (Guenther et al., 2006, 2012). The largest uncertainty stems from the base emission rates, which are extrapolated from very limited observations (Arneth et

al., 2008). PFT distributions are an additional source of uncertainty, with different land-cover maps producing as much as a factor of two difference in isoprene emissions (Millet et al., 2008). Environmental factor dependences are better understood, the dominant factor of variability being temperature (Palmer et al., 2006). Isoprene emissions can undergo large change over decadal scales in response to changing land cover (Purves et al., 2004; Zhu et al., 2017a).

Satellite observations of formaldehyde atmospheric columns provide "top-down" constraints on isoprene emissions to test inventories (Palmer et al., 2003, 2006; Millet et al., 2008; Barkley et al., 2013; Marais et al., 2014). HCHO is formed promptly and in high yield from isoprene oxidation, at least when concentrations of nitrogen oxides ($NO_x \equiv NO + NO_2$)



originating from combustion or soils are relatively high (Wolfe et al., 2016). A common approach has been to assume a local linear relationship between HCHO columns and isoprene emissions (Palmer et al., 2003, 2006; Millet et al., 2008), but this does not capture the spatial offset between the point of emission and the resulting HCHO column. This spatial offset can be hundreds of km, depending in particular on $NO_x$ levels (Marais et al., 2012). Tracing the observed HCHO back to the

5   location of isoprene emission requires accounting for this coupling between chemistry and transport. Previous studies have applied adjoint-based global inversions to account for transport in the isoprene-HCHO source-receptor relationship (Stavrakou et al., 2009; Fortems-Cheiney et al., 2012; Stavrakou et al., 2015; Bauwens et al., 2016), but they used older chemical mechanisms and horizontal resolutions of hundreds of km that do not capture the chemical time scales for isoprene conversion to HCHO.

Here we apply the adjoint of the GEOS-Chem chemistry-transport model at $0.25° \times 0.3125°$ horizontal resolution in an inversion of OMI HCHO observations to infer isoprene emissions in the Southeast US during the summer of 2013. Our inversion takes advantage of extensive aircraft observations of chemical composition from the NASA SEAC[4]RS campaign (Toon et al., 2016). These observations corrected and validated the OMI HCHO retrievals (Zhu et al., 2016), and allowed a

thorough evaluation of isoprene and $NO_x$ chemistry in GEOS-Chem including HCHO yields from isoprene oxidation as a function of $NO_x$ and time (Travis et al. 2016; Fisher et al., 2016; Chan Miller et al., 2017). They further showed that the $0.25° \times 0.3125°$ resolution of GEOS-Chem captures the spatial segregation between isoprene and $NO_x$ emissions that would be lost at coarser model resolution and introduce error in the HCHO yield (Yu, K. et al. 2016). The SEAC[4]RS observations provide unprecedented testbed for determining the value of satellite HCHO observations to quantify isoprene emissions on

ecosystem-relevant scales.

## 2 Methods

### 2.1 OMI observations

We use the OMI-SAO v003 Level 2 HCHO data as described by González Abad et al. (2015). The OMI spectrometer flies aboard the NASA Aura research satellite and provides daily global mapping with a local overpass time of 1330 and a nadir

resolution of $24 \times 13$ km$^2$. Slant column densities (SCD, $\Omega_s$) of HCHO are calculated by direct fitting of OMI radiances. The SCD over a remote Pacific reference sector is subtracted to give the enhancement over the background ($\Delta\Omega_s$). The SCD is related to the vertical column density (VCD, $\Omega$) by an air mass factor (AMF), which accounts for the sensitivity of the backscattered radiances to the HCHO vertical profile. The final VCD is calculated by adding the background VCD ($\Omega_o$) from the GEOS-Chem simulation over the Pacific reference sector:

$$\Omega = \frac{\Delta\Omega_s}{AMF} + \Omega_o \qquad (1)$$

The background contribution averages $3.8 \times 10^{15}$ molecules cm$^{-2}$, small relative to the enhancements over the Southeast US.



Zhu et al. (2016) validated the OMI-SAO v003 HCHO VCD satellite data during SEAC[4]RS by comparison to two independent in situ HCHO measurements aboard the aircraft. This verified the accuracy of the spatial and temporal patterns in the satellite data but revealed a 37% low bias, which was attributed to errors in spectral fitting and in assumed surface reflectivity. Following the recommendation of Zhu et al. (2016), we correct this bias by applying a uniform scaling factor of 1.59 to the satellite data. Independent evaluation with ground-based HCHO observations provides support for this correction factor (Zhu et al., 2017b).

Simulation of the OMI data with the GEOS-Chem model requires that we use an AMF consistent with the model vertical profile when converting observed SCDs to VCDs (or equivalently when converting model VCDs to SCDs). Here we calculate the AMF by applying the local OMI scattering weights from the operational retrieval to the GEOS-Chem HCHO vertical profile (Qu et al., 2017). The satellite data are filtered by the OMI-SAO quality flag, cloud fractions less than 0.3, solar zenith angles less than 60$^o$, and values within the range -0.5 to $10 \times 10^{16}$ molecules cm$^{-2}$ (Zhu et al., 2016). We accumulate 192,889 individual scenes over the 8-week period with an average of 35 single-scene observations per 0.25$^o$ × 0.3125$^o$ grid cell.

Single-scene measurement error includes (1) the spectral fitting error reported as part of the operational product, and (2) the error in the AMF calculation, which increases from 15% under clear sky conditions to 20% at a cloud fraction of 0.3 (Millet et al, 2006). We increase the spectral fitting error by a factor of 1.59, the same factor used to correct the mean bias in OMI VCDs. If the conversion of radiances to HCHO columns is the cause of the bias, we would expect this bias to translate to the spectral fitting error. This assumption is tested in section 3.3. Spectral fitting dominates the error budget, so that individual retrievals typically have an 80% error over the Southeast US. This error decreases when averaging over a large number of retrievals (Boeke et al., 2011).

Figure 1 shows the error-weighted mean OMI HCHO VCD during the August-September 2013 SEAC[4]RS period on the 0.25 × 0.3125° GEOS-Chem grid. The regional enhancement over the Southeast US is well known to be due to isoprene emission (Abbott et al., 2003; Palmer et al., 2003, 2006; Millet et al., 2006, 2008). The location of the maximum varies from year to year depending on temperature (Palmer et al., 2006).

## 2.2 MEGAN emissions

We use as prior estimate of isoprene emission the MEGAN v2.1 inventory (Guenther et al., 2012), as implemented in GEOS-Chem by Hu et al. (2015a). Base emission factors (top left panel of Fig. 2) are taken from the MEGAN v2.2 land cover map and correspond to emissions under standard conditions (temperature of 303 K, leaf area index=5, canopy 80% mature, 10%, old and 10% growing, and photosynthetic photon flux density of ~1500 µmol m$^{-2}$s$^{-1}$ at the canopy top). MEGAN v2.2 land



cover was constructed for the year 2008 based on the National Landcover Dataset (NLCD, Homer et al., 2004) and vegetation speciation from the Forest Inventory and Analysis (FIA, www.fia.fs.fed.us). It uses the 16-PFT classification scheme of the Community Land Model 4 (CLM4) and further specifies regionally variable base emission factors based on speciation. For example, the PFT base emission factor for the "Broadleaf Deciduous Temperate Tree" category varies

depending on the relative abundance of isoprene emitters (e.g., oak) and non-emitters (e.g., maple). The highest base emission factors are in the Ozarks of Southeast Missouri where pine-oak forests dominate the land cover (Wiedinmyer et al., 2005).

Actual isoprene emissions are computed locally by multiplying the base emission factors by environmental factors to

account for local conditions of leaf area index and leaf age, derived from MODIS observations (Myneni et al., 2007), and temperature and direct and diffuse solar radiation, taken from the GEOS-FP assimilated meteorological data used to drive GEOS-Chem. The resulting emissions are shown in the top right panel of Figure 2. The pattern differs from the base emission factors, primarily because of temperature. The highest emissions are in Louisiana and Arkansas, where temperatures are particularly high. The general spatial patterns of OMI HCHO (Fig. 1) and MEGAN v2.1 emissions show

broad similarities but also substantial differences. For example, OMI shows no enhancement over the Edwards Plateau in Texas where MEGAN v2.1 predicts high isoprene emissions. These differences will be analyzed quantitatively in our inversion.

**2.3 GEOS-Chem and its adjoint**

We use the GEOS-Chem chemical transport model and its adjoint (Henze et al., 2007), driven by assimilated NASA GEOS-

FP meteorological data in a nested configuration at $0.25° \times 0.3125°$ horizontal resolution (Zhang et al., 2015, 2016; Kim et al., 2015). Our model domain covers the Southeast US (102.812-77.188°W, 28.75-42.25°N; Fig. 1), taking initial and dynamic boundary conditions from a global simulation with $4° \times 5°$ horizontal resolution. We simulate an 8-week period (1 August – 25 September 2013) at the $0.25° \times 0.3125°$ horizontal resolution.

The GEOS-Chem adjoint version is v35k, which is based on version v8 of GEOS-Chem with updates through v9 (http://acmg.seas.harvard.edu/geos). Here we update the chemical mechanism in v35k to GEOS-Chem v9.02 (Mao et al., 2010, 2013) and further update isoprene chemistry as described by Fisher et al. (2016) and Travis et al. (2016) in their simulation of SEAC[4]RS observations. These updates include in particular (1) explicit representation of isoprene peroxy radical (ISOPO$_2$) isomerization and subsequent hydroperoxy-aldehyde (HPALD) formation, (2) formation of isoprene

epoxides (IEPOX) and their oxidation, and (3) a 24% increase in the HCHO yield from reaction of ISOPO$_2$ with NO. The updated oxidation mechanism better reproduces the time- and NO$_x$-dependence of HCHO production in the fully-explicit Master Chemical Mechanism v3.3.1 (Jenkin et al., 2015) and agrees with the HCHO yields derived from SEAC[4]RS and SENEX aircraft measurements over the Southeast US (Wolfe et al., 2016; Chan Miller et al., 2017; Marvin et al., 2017).

US anthropogenic emissions in GEOS-Chem are from the 2011 National Emissions Inventory (NEI11) of the US Environmental Protection Agency, scaled to 2013 (NEI, 2015). We decrease mobile $NO_x$ emissions by 60% from that inventory, as shown to be necessary to reproduce SEAC[4]RS and other 2013 observations for $NO_x$ and its oxidation products

including OMI observations of $NO_2$ (Travis et al., 2016; Chan Miller et al., 2017). Fire emissions, lightning $NO_x$ emissions, soil $NO_x$ emissions, non-isoprene MEGAN emissions, and updates to deposition are as in Travis et al. (2016). GEOS-FP diagnosed mixing depths are reduced by 40% to better match aerosol lidar observations during SEAC[4]RS (Zhu et al., 2016).

## 2.4 Inversion approach

The state vector $\mathbf{x}$ to be optimized in the inversion consists of temporally invariant scaling factors on the $0.25^o \times 0.3125^o$

GEOS-Chem grid applied to the prior MEGAN v2.1 isoprene emissions for the August-September 2013 SEAC[4]RS period. It consists of 4138 elements covering the land grid cells of the domain in Figure 1. Zhu et al. (2016) previously found that decreasing MEGAN v2.1 emissions by 15% improved the simulation of SEAC[4]RS HCHO observations and we include this correction in our prior estimate. Non-isoprene sources of HCHO do not contribute significantly to the HCHO column enhancements over the Southeast US (Millet et al., 2006; Zhu et al., 2014) and hence are not optimized as part of the

inversion.

The observation vector $\mathbf{y}$ consists of daily OMI HCHO columns (VCDs) calculated from OMI SCDs and GEOS-Chem AMFs mapped onto the $0.25^o \times 0.3125^o$ GEOS-Chem grid. We relate $\mathbf{y}$ to $\mathbf{x}$ using GEOS-Chem, denoted as $\mathbf{F}$ and representing the forward model for the inversion:

$$\mathbf{y} = \mathbf{F}(\mathbf{x}) + \mathbf{\varepsilon_0} \qquad (2)$$

GEOS-Chem HCHO columns are sampled at the OMI overpass time and filtered according to the same requirements outlined in section 2.1. The observational error vector $\mathbf{\varepsilon_O}$ includes contributions from the forward model error, the representation error, and the measurement error (Brasseur and Jacob, 2017). The representation error can be neglected here because the GEOS-Chem resolution is commensurate with the size of OMI pixels, and the forward model error is expected

to be small compared to the ~80% measurement error for individual scenes. Thus we take the measurement error as given in Section 2.1 to represent the observational error. The resulting observational error standard deviation averages $1.5 \times 10^{16}$ molecules cm$^{-2}$ for the domain of the inversion.

Assuming Gaussian error distributions and applying Bayes' theorem to weigh the information from the observations and the

prior estimate, the solution to the optimization problem involves minimization of the cost function $J(\mathbf{x})$ (Brasseur and Jacob, 2017):

$$J(\mathbf{x}) = (\mathbf{x} - \mathbf{x_A})^T \mathbf{S_A^{-1}}(\mathbf{x} - \mathbf{x_A}) + (\mathbf{F}(\mathbf{x}) - \mathbf{y})^T \mathbf{S_O^{-1}}(\mathbf{F}(\mathbf{x}) - \mathbf{y}). \qquad (3)$$



where $\mathbf{x_A} = (0.85, \ldots 0.85)^T$ is the prior estimate for $\mathbf{x}$, $\mathbf{S_A}$ is the corresponding prior error covariance matrix, and $\mathbf{S_O} = E[\mathbf{\varepsilon_O}\mathbf{\varepsilon_O}^T]$ is the observational error covariance matrix. We construct the prior error covariance matrix $\mathbf{S_A}$ by assuming 100% uncertainty in bottom-up emissions with no spatial error correlation. The sensitivity of the inversion to our assumptions for $\mathbf{S_A}$ and $\mathbf{S_O}$ will be tested in what follows.

The adjoint-based inversion enables a computationally tractable solution to the minimization of the cost function (3) when the forward model is highly non-linear, as is the case here. Starting from $\mathbf{x_A}$ as a first guess, the GEOS-Chem adjoint model calculates the local gradient of the cost function ($\nabla J(\mathbf{x_A})$) and passes it through the L-BFGS-B algorithm (Byrd et al., 1995; Zhu et al., 1997) to determine a next guess $\mathbf{x_1}$. It then recomputes ($\nabla J(\mathbf{x_1})$) and so on until convergence to the optimal value.

Convergence is reached when the cost function decreases by less than 1% over three consecutive iterations.

### 2.5 Error analysis

We examined the sensitivity of the inversion results to different assumptions made regarding the specification of errors. In the first and all subsequent sensitivity analyses, we use the reported spectral fitting error in the operational retrieval without the factor of 1.59 increase. This gives an average observational error standard deviation of $0.9 \times 10^{16}$ molecules cm$^{-2}$, 40%

smaller than in the standard case.

Our assumed prior error estimate of 100% on the MEGAN v2.1 isoprene emissions in the standard inversion is deliberately large to allow for the possibility of emissions being misplaced on the 0.25$^o$×0.3125$^o$ grid. We conducted a sensitivity analysis with a 50% prior error estimate.

The prior errors in the standard inversion have no spatial error correlation (i.e., $\mathbf{S_A}$ is diagonal), but some error correlation may in fact be expected depending on the homogeneity of land cover types. To test this, we conducted a sensitivity simulation where the state vector $\mathbf{x}$ of emission scaling factors is not optimized on the 0.25$^o$×0.3125$^o$ grid but instead on a coarser irregular grid defined using a hierarchical clustering algorithm (Johnson, 1967; Wecht et al., 2014) with geographical

proximity and commonality of MEGAN v2.1 emissions as clustering parameters. The resulting state vector is composed of 500 clusters, ~10 times fewer than the number of grid cells at 0.25$^o$ × 0.3125$^o$ resolution.

### 3 Results

### 3.1 Optimal estimate of isoprene emissions

Figure 2 shows optimized scaling factors for our standard inversion, and the resulting isoprene emissions (optimized

emissions = MEGAN emissions × scaling factors). Isoprene emissions are lower than MEGAN v2.1 by 40% on a regional average over the Southeast US domain, with decreases of more than a factor of 3 in some areas. Figure 3 summarizes the





results from the sensitivity analyses with different error assumptions. The standard inversion and the different sensitivity analyses show similar spatial patterns for emissions, with correlation coefficients $r$ = 0.96-0.98 on the $0.25^{\circ} \times 0.3125^{\circ}$ grid. The decrease in total regional emissions relative to MEGAN v2.1 ranges between 40% and 54%. The cluster inversion shows the largest decrease, because the smaller-dimension state vector allows for stronger fit from observations. However, aggregation errors in that inversion could cause overfit (Turner and Jacob, 2015).

Figure 4 shows the simulated HCHO columns from GEOS-Chem using the MEGAN v2.1 emissions and using the optimized emissions from the standard inversion (Fig. 3). The positive bias over high isoprene emitting regions using MEGAN v2.1 disappears when using the optimized emissions. The negative bias ($-3 \times 10^{15}$ molecules cm$^{-2}$) that persists over low-isoprene emitting regions is not corrected due to the high error associated with the OMI observations, and the low isoprene emissions in those regions. We attribute this to a bias in the background, unrelated to isoprene emission. The best agreement between OMI and GEOS-Chem is provided by the standard inversion configuration, as shown in Figs. 2 and 4. The standard inversion also provides the best agreement with SEAC$^4$RS data, as presented below.

## 3.2 Comparisons with SEAC$^4$RS data

In situ measurements of isoprene and its oxidation products aboard the SEAC$^4$RS aircraft provide an independent test of the inversion results. HCHO was measured in SEAC$^4$RS by two different techniques: mid-IR absorption spectroscopy using the CAMS (Richter et al., 2015), and laser-induced fluorescence using the NASA GSFC ISAF (Cazorla et al., 2015). The two measurements are well correlated ($r$ = 0.96 in the mixed layer), with ISAF ~10% higher than CAMS measurements (Zhu et al., 2016). Here we use the CAMS measurements, as these measurements were used in the validation of the OMI SAO product (Zhu et al., 2016). The associated measurement uncertainty is 4%. Isoprene and the sum of methyl vinyl ketone and methacrolein (MVK + MACR) were measured by PTR-MS (deGouw and Warneke, 2007), with reported uncertainties of 5% and 10%, respectively. Isoprene hydroperoxides (ISOPOOH) and isoprene nitrates (ISOPN) were measured by the Caltech CIMS (Crounse et al., 2006; Paulot et al., 2009a; St Clair et al., 2010), with respective uncertainties of 30 ppt + 40% and 10 ppt + 30%.

MVK+MACR measurements are corrected to account for the interference caused by the degradation of ISOPOOH on instrument surfaces (Rivera-Rios et al., 2014). The correction is calculated as MVK+MACR$_{corrected}$= MVK+MACR$_{measured}$ − X × ISOPOOH$_{measured}$, where X = 0.44 with a relatively large uncertainty of +0.21/-0.12. Formaldehyde measurements may suffer from a similar, but smaller interference. In the ISAF instrument, conversion of ISOPOOH to HCHO contributes negligibly (<4%) to the observed signal in ISOPOOH- and HCHO-rich environments, but a delay in ISOPOOH conversion and a rapid transition in sampling environments can manifest in more substantial (<10%) interferences (St. Clair et al., 2016). This has not yet been examined for the CAMS instrument.



We exclude data influenced by urban plumes ($[NO_2] > 4$ ppb), open fire plumes ($[CH_3CN] > 200$ ppt), and stratospheric air ($[O_3]/[CO] > 1.25$ mol mol$^{-1}$), and focus solely on measurements within the daytime boundary layer (0900-1800LT, <1.5km). In all comparisons with model results, observations are averaged over the GEOS-Chem grid at 10 min time steps.

Figure 5 shows the distribution of SEAC[4]RS observations. The aircraft flew over the Ozarks on several hot days, leading to the particularly high concentrations of isoprene and its oxidation products in the region. ISOPOOH is produced by the low-$NO_x$ pathway for isoprene oxidation, while ISOPN is produced by the high-$NO_x$ pathway. MVK and MACR are produced mostly by the high-$NO_x$ pathway. The spatial patterns reflect the contributions of both pathways across the Southeast US (Travis et al., 2016). Formaldehyde is more distributed because of the time lag in HCHO production from isoprene emission (Chan Miller et al., 2017). The relatively low correlation between spatially averaged isoprene and formaldehyde ($r = 0.49$, Figure 6) illustrates the importance of accounting for transport in inversions of HCHO data to infer isoprene emissions.

Figure 7 compares observed mixing ratios for isoprene and its oxidation products to the values simulated by GEOS-Chem using either MEGAN v2.1 isoprene emissions or the optimal estimate from the inversion. MEGAN v2.1 emissions lead to a factor of 2.5 overestimate in SEAC[4]RS observations of isoprene and ISOPOOH, a 50% overestimate in HCHO, factor of 2 overestimate for MVK+MACR, and 20% overestimate for ISOPN. The optimal estimate decreases the simulated concentrations and produces agreement with all observations within measurement uncertainty. The effect on isoprene and ISOPOOH is particularly large because the correction of emissions is strongest in high-emitting regions, which happen to also have low $NO_x$ (Figure 6). The reduction in HCHO, MVK+MACR, and ISOPN is less pronounced. Zhu et al. (2016) previously found no net model bias relative to the SEAC[4]RS HCHO observations using MEGAN v2.1 emissions reduced by a uniform 15%, but they used a lower HCHO yield from the $ISOPO_2$ + NO reaction and did not compare to other species. Travis et al. (2016) previously reported a factor of 2 overestimate of ISOPOOH in their SEAC[4]RS simulation with MEGAN v2.1 reduced by 15%, and the lower emissions in our optimal estimate effectively correct that bias.

Much of the residual scatter in the comparison of simulated vs. observed HCHO using optimized isoprene emissions appears to be caused by local bias in $NO_x$ (Figure 8). There is no mean $NO_x$ bias in our inversion (Travis et al., 2016) but there can be local bias. We find that local model biases in simulating HCHO observations are strongly correlated with corresponding model errors in $NO_x$, reflecting the $NO_x$-dependence of HCHO production from isoprene (Figure 6 and Chan Miller et al., 2017). When excluding points with more than 50% error in $NO_x$, the correlation between measured and simulated HCHO improves from $r = 0.62$ to $r = 0.70$ (n = 1222 to n = 708). This emphasizes the importance for inversions of HCHO data to use unbiased $NO_x$ concentrations.



## 4 Implications for isoprene emission inventories

Our results indicate that MEGAN v2.1 isoprene emissions over the Southeast US should be decreased by an average of 40%, consistent with previous analyses of OMI HCHO data that inferred 25-50% decreases (Millet et al., 2008; Bauwens et al., 2016). MEGAN v2.1 isoprene emissions are typically ~50% higher than the emissions calculated from the BEIS3 inventory
often used in US air quality models (Warneke et al., 2010; Carlton and Baker, 2011). BEIS and MEGAN both follow the emissions algorithms outlined in Guenther et al. (2006), but they use different canopy models and base emission factors (Bash et al., 2016). The geographic specificity of our high-resolution inversion allows us to examine potential causes of the MEGAN v2.1 overestimate in various environments. Below, we discuss three ecoregions in greater detail.

The high base isoprene emission factors in the Ozarks ecoregion (Figure 2) have led this region to be dubbed the "isoprene volcano" (Wiedinmyer et al., 2005). We find a 46% reduction in emissions in the region relative to MEGAN v2.1, in good agreement with a SEAC⁴RS estimate derived from isoprene flux profiles (Wolfe et al., 2015). Independent aircraft measurements over the Southeast US during the summer of 2013 found that MEGAN v2.1 was biased high by a factor of two for mixed pine-oak forests that are typical of the Ozarks (Yu, H. et al., 2017). These authors suggest that non-emitting
trees in the upper canopy may shade emitting trees, leading to lower than anticipated isoprene emissions.

The hot spot of isoprene emissions in the South Central Plains (Figure 2) is also reduced by 48% in our inversion relative to MEGAN v2.1. This region is dominated by needle leaf trees, with isoprene emissions stemming from the sweetgum/tupelo understory. Again, vertical heterogeneity or an incorrect fraction of emitters could lead to the MEGAN overestimate of
emissions. Alternatively, the base emission factor of sweetgum and tupelo could be significantly less than the assigned MEGAN value.

The Edwards Plateau in central Texas is a major isoprene source region in MEGAN v2.1, with base emission factors as high as in the Ozarks (Fig. 2), but our inversion decreases emissions in that region by more than a factor of three. In contrast, a
land cover map used for BEIS (BELD4) shows no hotspot (Wang et al., 2017). Both land cover maps are derived from the NLCD, but they follow different methodologies for translating NLCD classifications to base emission factors. Land cover estimates vary widely for this region. NLCD-based maps show the Edwards Plateau dominated by broadleaf trees, whereas the MODIS land cover product is dominated by grasses, leading to a factor of 10 lower isoprene emissions (Huang et al., 2015). Given the wide range of land-cover and emission factor estimates, a better understanding of land-cover is needed to
probe the causes of bias in this region.



## 5 Implications for surface air quality

Isoprene emissions can either increase or decrease surface ozone in air quality models, depending on the local chemical environment and the mechanism used (Mao et al., 2013). We find in GEOS-Chem that our optimized isoprene emissions lead to a decrease in mean surface afternoon $O_3$ concentrations by ~1-3 ppb over the Southeast US relative to the standard
simulation using MEGAN v2.1 emissions. The GEOS-Chem simulation of Travis et al. (2016) previously found an 8 ppb overestimate of surface ozone over the Southeast US during SEAC[4]RS, which a subsequent analysis by Travis et al. (2017) attributed in part to unresolved surface layer gradients and an overestimate of vertical mixing; we find here that the overestimate of isoprene emissions could also contribute.

Isoprene is also a precursor for organic aerosol (OA), which is a dominant contributor to fine particulate matter ($PM_{2.5}$) in surface air (Zhang et al., 2007). In a previous GEOS-Chem simulation of the SEAC[4]RS period, Kim et al. (2015) found that isoprene contributes 40% of total OA over the Southeast US in summer, assuming a 3% mass yield from isoprene oxidation and MEGAN v2.1 isoprene emissions reduced by 15%. A more mechanistic study of OA formation from isoprene oxidation under the SEAC[4]RS conditions found a 3.3% mass yield, most of which was produced in the low-$NO_x$ pathway (Marais et
al., 2016). Our work finds a factor of 2 decrease in ISOPOOH relative to the simulation using MEGAN v2.1 emissions reduced by 15%, and consistent with observations (Fig. 6). This suggests that isoprene OA formation may be only half of the value found by Kim et al. (2015), implying that other sources such as terpenes may make more important contributions to OA (Pye et al., 2010, 2015; Xu et al., 2015).

## 6 Conclusions

We used newly validated HCHO observations from the OMI satellite instrument to demonstrate the capability for applying these satellite observations to fine-resolution inversion of isoprene emissions from vegetation. Our work focused on the Southeast US where aircraft observations from the NASA SEAC[4]RS campaign provide detailed chemical information on isoprene and its oxidation products (including HCHO) to independently evaluate the inversion. The inversion used the adjoint of the GEOS-Chem chemical transport model at 0.25° × 0.3125° horizontal resolution and leveraged on previous
studies that applied GEOS-Chem to simulation of the SEAC[4]RS observations including in particular for $NO_x$. HCHO yields from isoprene oxidation are highly sensitive to $NO_x$ levels, and the high resolution of the GEOS-Chem inversion allowed us to properly describe the spatial segregation between isoprene and $NO_x$ emissions.

We found that the MEGAN v2.1 inventory of isoprene emissions commonly used in atmospheric chemistry models is biased
high on average by 40% across the Southeast US. This is consistent with several previous top-down studies and recent analyses using flight-based flux and eddy covariance measurements. Our optimized emissions produce better agreement with SEAC[4]RS observations of isoprene and its oxidation products including HCHO. Local model errors in simulating HCHO



observations along the aircraft flight tracks are highly correlated with local model errors in $NO_x$. This highlights the importance of accurate $NO_x$ fields in inversions of HCHO observations to infer isoprene emissions.

The high resolution of our inversion allows us to quantify isoprene emissions and analyze MEGAN v2.1 biases on
ecosystem-relevant scales. We find that MEGAN v2.1 is biased high everywhere across the Southeast US but is correct in placing maximum 2013 emissions in Arkansas/Louisiana/Mississippi. The Ozarks Plateau in Southeast Missouri has particularly high base emission factors in MEGAN v2.1, reflecting the abundance of oak trees, but isoprene emissions there are dampened by relatively low temperatures and our results further suggest an overestimate in the base emission factors. Another prominent overestimate is over the Edwards Plateau in central Texas, where MEGAN v2.1 emissions are biased
high by a factor of 3 according to our inversion possibly reflecting errors in land cover. Our results suggest that the BEIS inventory may yield more accurate isoprene emissions for these areas.

Our downward correction of isoprene emissions in GEOS-Chem as a result of the inversion leads to a 1-3 ppb reduction in modeled surface $O_3$, correcting some of the overestimate previously found in the model. It also decreases the contribution of
isoprene to organic aerosol, possibly suggesting a greater role for terpenes.

HCHO observations from space are expected to improve considerably in the near future. TROPOMI, launched in October 2017, will provide global HCHO and $NO_2$ observations at 7 km × 7 km nadir resolution daily (Veefkind et al., 2012), as compared to 24 km x13 km for OMI. Concurrent HCHO and $NO_2$ observations can provide a check against model bias in
$NO_x$ affecting the yield of HCHO from isoprene (Marais et al., 2012). The TEMPO geostationary instrument to be launched in the 2019-2022 window will provide HCHO and $NO_2$ observations at 2 km × 4.5 km pixel resolution multiple times per day (Zoogman et al., 2017). Coupled with the high-resolution inversion framework shown here, these future observations may greatly improve our ability to quantify US isoprene emissions from space.

## 7 Data availability

The OMI-SAO Version-3 Formaldehyde Product is available at the NASA Goddard Earth Sciences Data and Information Services Center (https://aura.gesdisc.eosdis.nasa.gov/data/Aura_OMI_Level2/OMHCHO.003/). SEAC[4]RS observations are available from the NASA LaRC Airborne Science Data for Atmospheric Composition (https://www-air.larc.nasa.gov/cgi-bin/ArcView/seac4rs, doi:10.5067/Aircraft/SEAC4RS/Aerosol-TraceGas-Cloud). The adjoint of the GEOS-Chem model is available at http://wiki.seas.harvard.edu/geos-chem/index.php/GEOS-Chem_Adjoint.

*Acknowledgements.* We are grateful for the contributions from all members of the SEAC[4]RS flight and science teams. We acknowledge Thomas B. Ryerson for his contribution of the $NO_X$ measurements. Tomas Mikoviny is acknowledged for his



support with the PTR-MS data acquisition and analysis. PTR-MS measurements during SEAC4RS were supported by the Austrian Federal Ministry for Transport, Innovation and Technology (bmvit) through the Austrian Space Applications Programme (ASAP) of the Austrian Research Promotion Agency (FFG). Funding was provided by the NASA Aura Science Team.



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



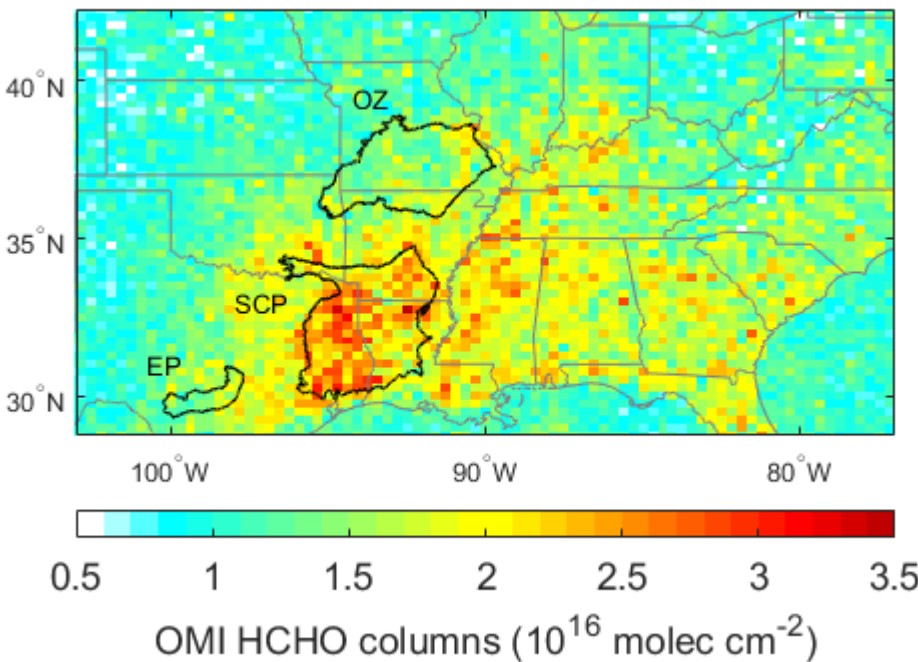

**Figure 1: Error-weighted mean OMI HCHO vertical column densities for the SEAC$^4$RS time period (1 Aug 2013 – 25 Sept 2013). The Edwards Plateau (EP), Ozarks (OZ), and South Central Plains (SCP) ecoregions are denoted with black outlines (https://www.epa.gov/eco-research/ecoregions, level 3 and 4 data).**



**Figure 2: Isoprene emissions in the Southeast US. Top: MEGAN v2.1 base isoprene emission factors and emissions for the SEAC⁴RS time period. Bottom: Scaling factors from the inversion and optimized emissions. The color scale differs for MEGAN and optimized emissions. The Edwards Plateau (EP), Ozarks (OZ), and South Central Plains (SCP) ecoregions are denoted with black outlines.**



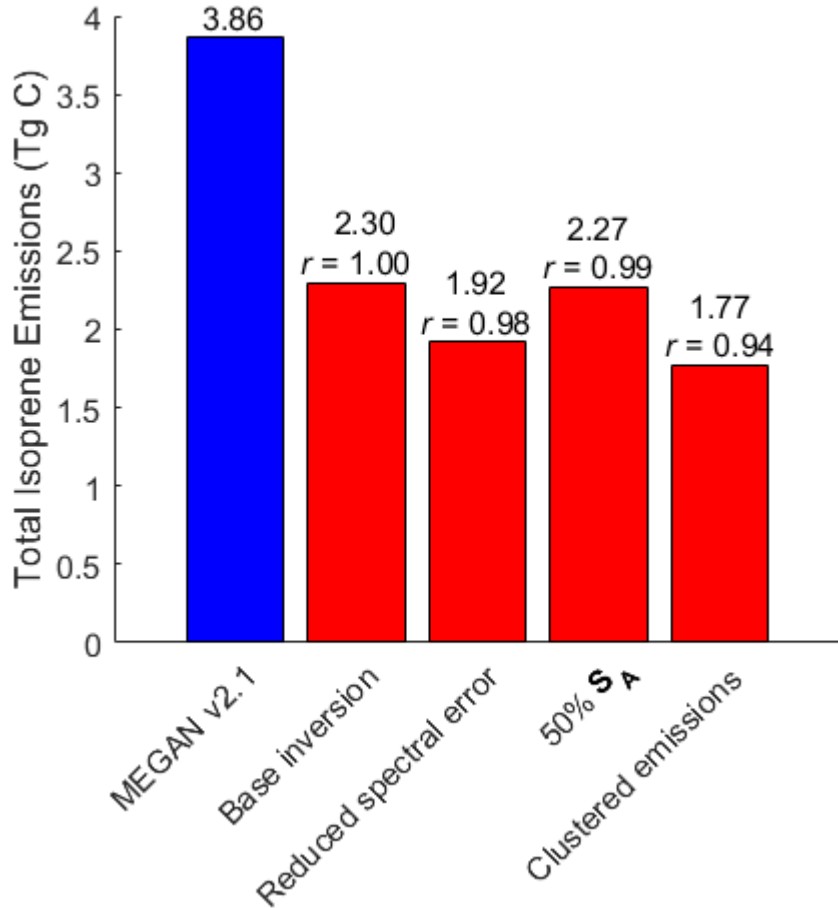

**Figure 3: Total isoprene emissions for the Southeast US domain of Figure 1 over the period 1 Aug-25 Sept 2013. The MEGAN v2.1 inventory value is compared to results from the base inversion applied to the OMI formaldehyde data (optimized emissions in Figure 2) and to sensitivity inversions using different error specifications (see text for details). Numbers on top of each bar are the total isoprene emissions, and correlation coefficients (*r*) describe the spatial consistency between the base inversion (*r* = 1) and the sensitivity inversions.**



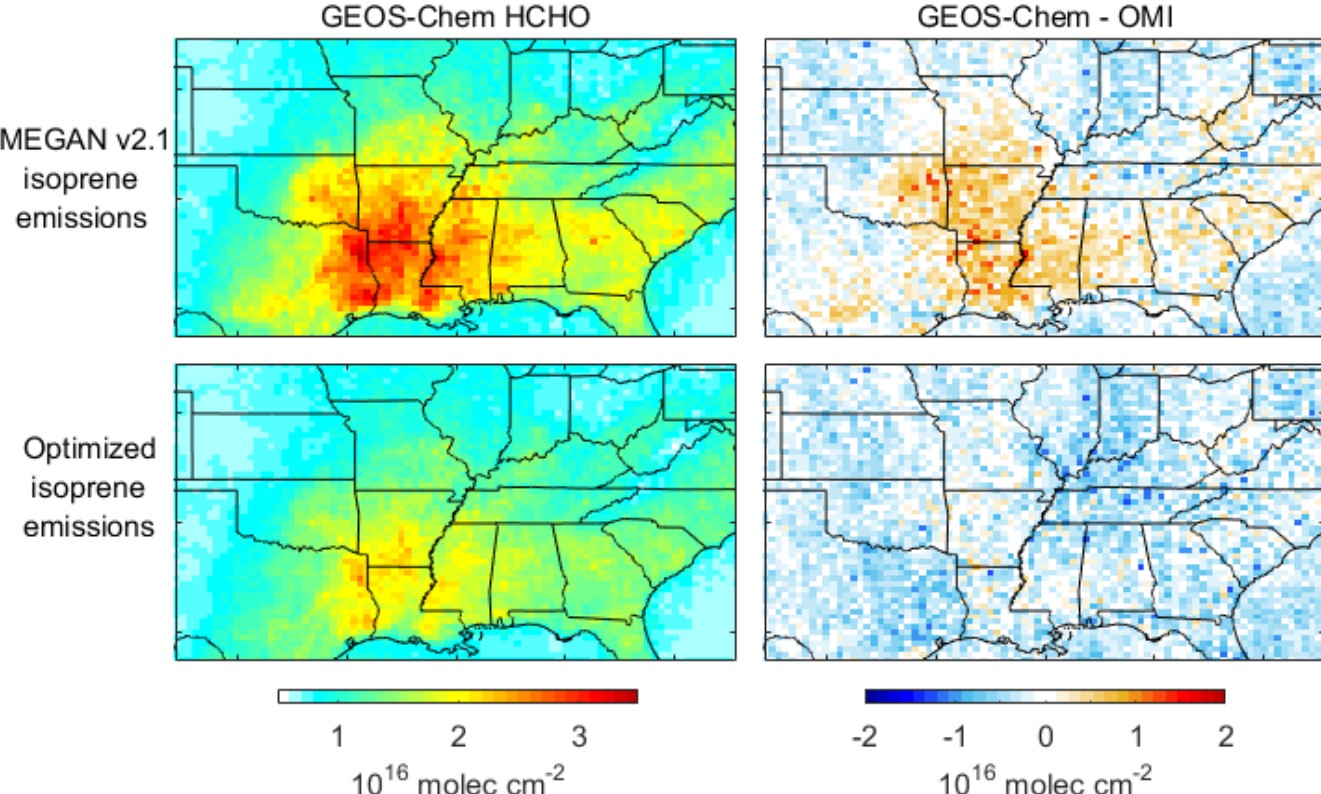

**Figure 4: Simulated HCHO vertical column densities and model bias using prior and optimized isoprene emissions. Values are averages for 1 Aug – 25 Sep 2013 at the OMI overpass time (1330 local), weighted by the OMI measurement error as in Figure 1. The right panels show the differences between the simulated columns and the OMI observations from Figure 1.**




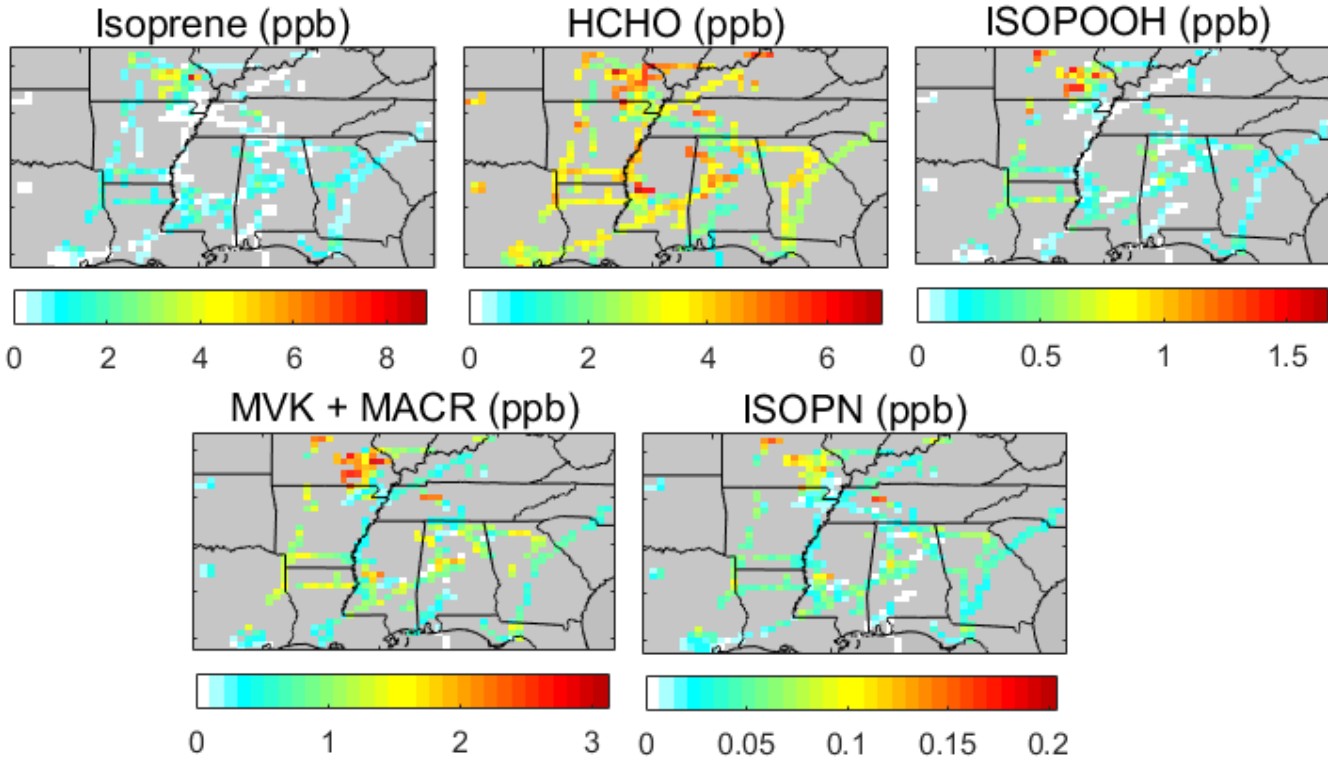

**Figure 5: Mean boundary layer concentrations of isoprene and its oxidation products measured in the SEAC[4]RS aircraft campaign (1 Aug – 25 Sep 2013). The observations are for daytime (0900-1800 LT) below 1.5 km altitude, and exclude urban and fire plumes as described in the text.**



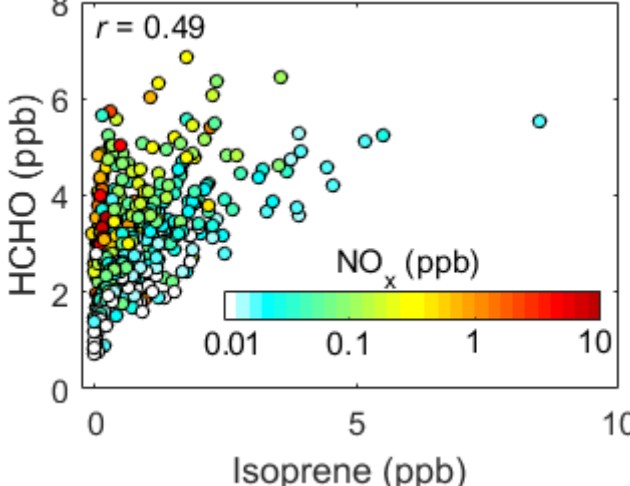

**Figure 6: Spatially averaged concentrations of boundary layer isoprene and HCHO measured during SEAC[4]RS, colored by observed NO$_x$. Data are the same as in Figure 5.**





**Figure 7: Comparison of SEAC⁴RS observations and modeled mixing ratios using either MEGAN v2.1 (blue) or the optimized isoprene emissions (red) from the base inversion of OMI HCHO data (Figure 2). The dashed line indicates 1:1 agreement. The colored lines are the reduced major axis linear regressions and the inset numbers are the corresponding slopes, with error standard deviations inferred from bootstrap sampling.**

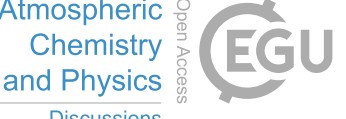

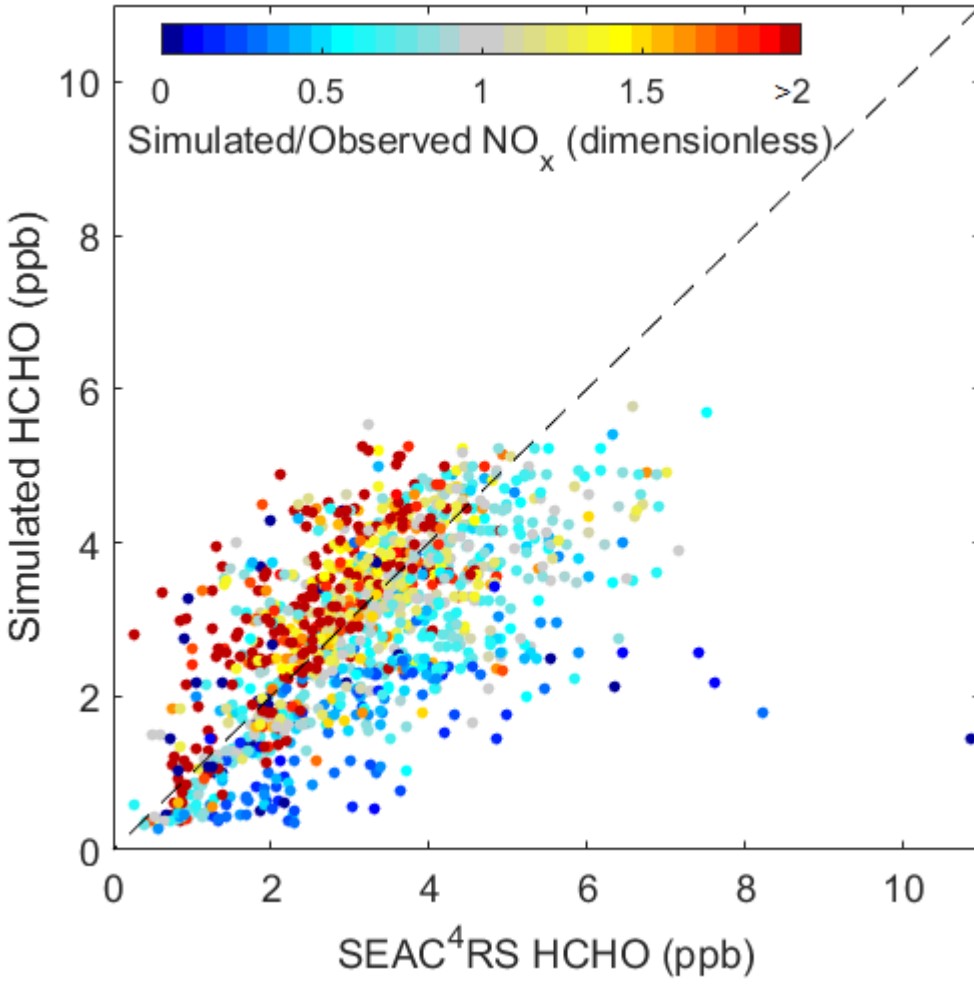

**Figure 8: Comparison of simulated and observed HCHO along the SEAC$^4$RS flight tracks, using the model with optimized isoprene emissions from the base inversion. The dashed line indicates 1:1 agreement. Data are the same as in Figure 7 (upper middle) but are colored by the local ratio of simulated to observed NO$_x$ concentration.**