# Peer review of "High-resolution inversion of OMI formaldehyde columns to quantify isoprene emission on ecosystem-relevant scales: application to the Southeast US"

_Atmospheric Chemistry and Physics, 2017_

## Referee Comment (RC1) · Anonymous Referee #1 · 21 Jan 2018

The authors present an adjoint inversion of isoprene emissions over the SE US based on OMI HCHO observations. The results are evaluated on the basis of independent aircraft data and interpreted in terms of their implications for our understanding of isoprene emission drivers, and for regional ozone and PM.

Overall the analysis is well-done, and provides a useful and interesting addition to the literature in this area. Analysis techniques are state-of-the-art, and the topic is germane

to ACP. I have some comments and suggestions for improving the manuscript below. Once these are addressed the paper merits publication in ACP.

Science comments.

The sensitivity / error analysis only addresses assumptions for the error covariance matrices, and also a clustering approach for optimization. Model errors are not addressed. This gravity of this is lessened by the fact that many of the key variables have been separately evaluated with SEAC4RS data in prior publications. However, remaining errors in these parameters can still be expected to affect your results. A more thoughtful error analysis should be done, and would make your results more convincing. For example, a more comprehensive set of sensitivity inversions with altered assumptions for key variables (e.g., model NOx, temperature, mixing heights, OMI cloud threshold, . . .).

How accurate are the GEOS-FP temperatures over these regions? What model temperature (skin, surface, 2m, lowest-box) is used to compute emissions? Are we sure that meteorological biases are not a significant part of the discrepancies you see?

8-11. " We attribute this to a bias in the background". This implies that your background correction approach is not working properly. The background bias would then also apply to the high-isoprene areas. Since isoprene gives rise to an HCHO enhancement on top of that background, wouldn't this mean that your downward isoprene adjustments should be even larger? I.e. if the blue color everywhere in Fig 4 is really due to a background bias unrelated to isoprene, then surely if the isoprene adjustments were physically correct one would expect the same blue color throughout Figure 4 (bottom-right panel). Visually it appears that the high-isoprene areas average to ∼zero in the a posteriori, but this is not the case elsewhere in the domain.

Figure 4. It is interesting that while the inversion clearly improves agreement over the high-isoprene areas, it appears to make the low-bias worse in adjoining areas like Tennessee/Kentucky and east Texas. Posteriori biases there look to be larger than the generic background bias elsewhere. What do you make of this? Does this imply an

[Figure]

over-correction of the isoprene emissions, or some spatial mis-representation of the isoprene-HCHO conversion?

Fig 7- The improved agreement with respect to (nearly) all the related SEAC4RS tracers is a very nice result. Do you attribute any significance to the fact that agreement worsened somewhat for ISOPN?

2-10, where is this 1/3 estimate coming from?

2-24, "the largest uncertainty stems from the base emission rates". I don't think this is necessarily categorically true anymore. Certainly it will depend on the location and spatial scale being examined. If the land cover is wrong (are there oak trees or not, for example) that will give a very large emission error. Assimilated meteorological fields are frequently wrong by a degree or 3, which again can cause major emission biases. In some cases and places I'm sure you're right that emission capacities are the biggest source of error but I disagree that is always the case.

2-26, yes, the environmental factor dependencies are fairly well understood but that doesn't mean a model has the temperature right, or for that matter the distribution of temperature through a plant canopy.

P9-L10 "The relatively low correlation between spatially averaged isoprene and formaldehyde (r = 0.49, Figure 6) illustrates the importance of accounting for transport in inversions of HCHO data to infer isoprene emissions." Sure, but this importance depends on the resolution at which one is attempting to compute emissions. At the resolution used here it is clearly quite important.

---

## Referee Comment (RC2) · Anonymous Referee #2 · 3 Feb 2018

Kaiser et al. use two months (August-September 2013) of formaldehyde column measurements from OMI to infer top-down isoprene emissions over the Southeast US. This region is well known for its high isoprene emissions and has been the subject of a large body of previous studies focusing on the estimation of isoprene fluxes using satellite and/or aircraft observations. Here, by using the adjoint of the GEOS-Chem CTM at high spatial resolution, the authors conclude that (i) the a priori inventory of isoprene emissions is biased high by 40% on average over the Southeast US, as was already pointed out in previous studies, (ii) the a priori inventory is likely too high by a factor of

3 in Central Texas, and (iii) the top-down isoprene estimates improve the model skill to reproduce observations of the SEAC[4]RS concurrent campaign. The study is interesting, the manuscript is well written and clearly structured. The overall results appear reasonable and are well discussed. I therefore recommend publication to Atmospheric Chemistry and Physics after the following questions are adequately addressed.

General comments :

- There seems to be a contradiction between the GEOS-Chem simulation in the present manuscript and in Zhu et al. (2016), regarding the comparison with the SEAC[4]RS CAMS data. In Zhu et al. (2016), GEOS-Chem HCHO columns had to be increased by 10% in order to match SEAC[4]RS CAMS data. This might be partly explained by the 15% reduction of MEGAN isoprene fluxes in that paper. But still, it is very surprising that the GEOS-Chem model is now found to overestimate HCHO by a factor of 1.47 compared to the same CAMS data. I have serious doubts about the fact that the 24% increase in HCHO yield in the $ISOPO_2$+NO reaction can lead to an overestimate of 47%, and if so, this should be demonstrated.

- p.3, l.7 : I suggest to remove 'older chemical mechanisms and' from the sentence. 'Old' is always relative and for example the latest findings in isoprene chemistry (Bates et al. 2016, Teng et al. 2017, etc.) are not considered in the present study.

- In Section 2.2, the effect of soil moisture stress on the bottom-up isoprene fluxes is not discussed. Have you accounted for it in your simulations? The Edwards Plateau in Central Texas is often affected by drought. The strong flux decrease derived in this region might be partly explained by the neglect of soil moisture stress in MEGAN. This warrants some discussion.

- p.6, l.2 : I'm a bit confused here. In Travis (2016), not only mobile NOx emissions are reduced by 60% but all non-power plant sources (or alternatively 30% reduc-
tion of non-power plant sources and no soil NO emissions). In Chan Miller et al. (2017) a decrease of 50% in anthropogenic NOx emissions relative to NEI 2011 is applied. Please clarify what you actually did in the present work and why. Note furthermore that soil NO emissions were found to be too low over the Ozarks by Wolfe et al. (2015).

- p. 6, l. 15 : Still, it would be useful to give percentage estimate of contribution of other NMVOCs to the formaldehyde columns in Southeast US.

- p.7, l. 14 : I don't see what is the motivation of the first sensitivity inversion (with reduced errors). Please explain.

- p.10, l.4 : 50%, do you mean factor of 2 or 1.5?

- Figure 1 : The OMI SAO columns over the SEAC$^4$RS period do not look the same as in Figure 5 of Zhu et al. (2016). What is the difference? Why did you use error-weighted means? Please specify how these means are calculated (relative or absolute).

- Figure 2 : The MEGAN base emission factors shown in the upper left panel should be the same as in Zhu et al. (2016) (both use Hu et al. 2015). They are however about 10 times higher that in Figure 3 of Zhu et al. (2016). Please explain.

References :

Bates, K. H. (2016), Production and fate of C4 dihydroxycarbonyl compounds from isoprene oxidation, J. Phys. Chem. A, 120(1), 106-117.

Hu, L. et al. (2015), Isoprene emissions and impacts over an ecological transition region in the US Upper Midwest inferred from tall tower measurements, J. Geophys. Res., 120, 3553–3571.

Teng, A. P., Crounse, J. D. Wennberg, P. (2017), Isoprene peroxy radical dynamics, J. Am. Chem. Soc., 139, 5367–5677.

Travis, K. R., et al. (2016), Why do models overestimate surface ozone in the Southeast United States? Atmos. Chem. Phys., 16, 13561–13577.

Wolfe, G. M., et al. (2015), Quantifying sources and sinks of reactive gases in the lower atmosphere using airborne flux observations, Geophys. Res. Lett., 42, 8231–8240, doi:10.1002/2015GL065839.

Zhu, L. et al. (2016), Observing atmospheric formaldehyde (HCHO) from space: validation and intercomparison of six retrievals from four satellites (OMI, GOME2A, GOME2B, OMPS) with SEAC[4]RS aircraft observations over the southeast US, Atmos. Chem. Phys., 16, 13477–13490.

---

## Author Comment (AC1) · 3 Apr 2018

We thank the reviewers for their careful reading and their helpful questions and remarks. Their comments are shown in black, and our responses are shown in blue. The revised manuscript follows.

**Reviewer 1**

The sensitivity / error analysis only addresses assumptions for the error covariance matrices, and also a clustering approach for optimization. Model errors are not addressed. This gravity of this is lessened by the fact that many of the key variables have been separately evaluated with SEAC4RS data in prior publications. However, remaining errors in these parameters can still be expected to affect your results. A more thoughtful error analysis should be done, and would make your results more convincing. For example, a more comprehensive set of sensitivity inversions with altered assumptions for key variables (e.g., model NOx, temperature, mixing heights, OMI cloud threshold, . . .).

Our understanding of the HCHO-isoprene relationship is supported by the extensive evaluation of modeled  $NO_x$ , temperature, and mixing heights provided in previous work, alongside validated OMI retrievals. Changing these parameters would take substantial effort, and provide either predictable or negligible differences. Specifically:

- Using uncorrected GEOS-FP mixing depths decreases modeled average midday HCHO columns by less than 10%. Because modeled vertical profiles are used to interpret OMI observations, we expect this to have a low impact on our inversion.
- Comparison with SEAC4RS observations allows us to correct for bias in GEOS-FP temperatures (discussed further in following point). An error of +1K would result in an 8% difference in the temperature correction factor, which is well below the correction derived from our inversion.
- Our results highlighted in Figure 8 demonstrate the sensitivity of modeled HCHO to NOx. Running multiple inversions at different NOx levels is beyond the scope of this work.
- Our OMI data filtering and correction is based on the validation of Zhu et al. (2016). Validation has not been performed for other filtering criteria.

We have included the following statement to section 2.5 (error analysis):

"A general assumption in Bayesian optimization is that observational errors are randomly distributed, as opposed to systematic bias. Previous analyses of SEAC4RS observations provide some confidence as to this lack of bias. The validation work of Zhu et al. (2016) led to removal of bias from the OMI HCHO satellite data. The work of Travis et al. (2016) and Fisher et al. (2016) removed bias in the GEOS-Chem simulation relating isoprene emission to HCHO production. GEOS-FP biases in temperature and mixing depths were corrected by Fisher et al. (2016) and Zhu et al. (2016), respectively. All of these corrections have been implemented in our simulation."

How accurate are the GEOS-FP temperatures over these regions? What model temperature (skin, surface, 2m, lowest-box) is used to compute emissions? Are we sure that meteorological biases are not a significant part of the discrepancies you see?

GEOS-FP temperatures were compared to temperatures measured during the SEAC4RS campaign. A small positive bias was observed at high temperatures, which was corrected in this and all previous SEAC4RS publications (for GEOS-FP T>293 K, corrected temperature (K) =  $0.792 \times (\text{GEOS-FP} + 76.5\text{K})$ ). We assume this bias also effect GEOS-FP skin temperatures used

to calculate MEGAN emissions, and use the same correction approach. We have added this description to the discussion of isoprene emissions (section 2.2):

**"GEOS-FP temperatures in the boundary layer averaged 1 K higher than the SEAC4RS observations, and a downward correction is applied to the skin temperatures used in the computation of isoprene emissions in GEOS-Chem."**

8-11. "We attribute this to a bias in the background". This implies that your background correction approach is not working properly. The background bias would then also apply to the high-isoprene areas. Since isoprene gives rise to an HCHO enhancement on top of that background, wouldn't this mean that your downward isoprene adjustments should be even larger? I.e. if the blue color everywhere in Fig 4 is really due to a background bias unrelated to isoprene, then surely if the isoprene adjustments were physically correct one would expect the same blue color throughout Figure 4 (bottom right panel). Visually it appears that the high-isoprene areas average to ~zero in the a posteriori, but this is not the case elsewhere in the domain.

We agree that this suggests the background correction approach for the OMI-SAO product may be insufficient. Zhu et al (2016) derive a 37% bias. A more accurate correction may involve a smaller multiplicative term and an offset (i.e, rather than Corrected =  $A \times OMI$ , Corrected =  $A' \times OMI + B$ ).

Regardless of the formulation, the corrected OMI columns would have the same value over the high emitting regions, which are the derived in the Zhu et al. (2016) paper. A two-term correction would therefore yield the same isoprene scaling for high emitting regions.

Figure 4. It is interesting that while the inversion clearly improves agreement over the high-isoprene areas, it appears to make the low-bias worse in adjoining areas like Tennessee/Kentucky and east Texas. Posteriori biases there look to be larger than the generic background bias elsewhere. What do you make of this? Does this imply an over-correction of the isoprene emissions, or some spatial mis-representation of the isoprene-HCHO conversion?

The reviewer is correct that posterior biases are worse in some regions such as Eastern Texas and Tennessee/Kentucky. This is likely an over-correction that is an artifact of two aspects of our optimization approach.

First, the prior estimate  $\mathbf{x}_a=0.85$  weighs on the inversion such that larger scaling factors ( $\mathbf{x}_a >>1$ ) are not explored even in regions where the prior yields a negative bias in HCHO columns.

Second, the relative term in the observational error could lead to an overcorrection. In each grid cell, the observational term of the cost function are calculated as

$$J_{obs}(x) = \frac{(\mathbf{y} - \mathbf{F}(x))^2}{\sigma_0^2}$$

where x is the scaling factor, y are observed HCHO columns,  $\mathbf{F}(x)$  are modeled HCHO columns, and  $\sigma_0^2$  is the observational error variance.

The figure to the right demonstrates the asymmetry in  $J_{obs}(x)$ , using the example of  $\sigma_0=0.20y+0.3$ , where 0.20 represents the relative error associated with the air mass factor and b

represents the spectral fitting error. Because a portion of  $\sigma_0$  is relative to y, there is a larger penalty associated with overestimating low values than underestimating higher values. That is, for a gridcell with large variability in observed HCHO columns, the inversion preferentially optimizes isoprene emissions to match lower observations.

Though the values we show in Figure 4 are weighted by the observational error, the cost function scales with the variance (error squared).

Fig 7- The improved agreement with respect to (nearly) all the related SEAC4RS tracers is a very nice result. Do you attribute any significance to the fact that agreement worsened somewhat for ISOPN?

The worsening agreement with modeled ISOPN is a product of a subset of measurements where measured ISOPN is greater than 100 ppt. For this group of points, modeled  $NO_x$  is underestimated. ISOPN is more sensitive to  $NO_x$  than HCHO, and these points weigh heavily on the regression.

2-10, where is this 1/3 estimate coming from?

We have clarified the references in this statement as follows:

**Isoprene from vegetation comprises about one third of the global emission of volatile organic compounds (VOCs) (Guenther et al., 2006). Emissions in the southeastern United States during summertime are some of the highest in the world (Guenther et al., 2012).**

2-24, "the largest uncertainty stems from the base emission rates". I don't think this is necessarily categorically true anymore. Certainly it will depend on the location and spatial scale being examined. If the land cover is wrong (are there oak trees or not, for example) that will give a very large emission error. Assimilated meteorological fields are frequently wrong by a degree or 3, which again can cause major emission biases. In some cases and places I'm sure you're right that emission capacities are the biggest source of error but I disagree that is always the case.

We agree that errors in assimilated meteorological fields can cause large errors in isoprene emissions. However, here we are discussing uncertainties in the construction of bottom-up inventories rather than errors that can arise during their use. This is now specified in the text:

**"The largest uncertainty in the construction of bottom-up inventories stems from the base emission rates"**

2-26, yes, the environmental factor dependencies are fairly well understood but that doesn't mean a model has the temperature right, or for that matter the distribution of temperature through a plant canopy.

The reviewer's point is well taken. We have added the following statement to the text:

**"Factor dependences on environmental variables are better understood, the dominant factor of variability being temperature (Palmer et al., 2006), though any uncertainties in temperature will propagate into uncertainties in isoprene emission estimates."**

P9-L10 "The relatively low correlation between spatially averaged isoprene and formaldehyde (r = 0.49, Figure 6) illustrates the importance of accounting for transport in inversions of HCHO data to infer isoprene emissions." Sure, but this importance depends on the resolution at which one is attempting to compute emissions. At the resolution used here it is clearly quite important.

We have edited this sentence to specify that this is important at high resolution.

**Reviewer 2:**

There seems to be a contradiction between the GEOS-Chem simulation in the present manuscript and in Zhu et al. (2016), regarding the comparison with the SEAC4RS CAMS data. In Zhu et al. (2016), GEOS-Chem HCHO columns had to be increased by 10% in order to match SEAC4RS CAMS data. This might be partly explained by the 15% reduction of MEGAN isoprene fluxes in that paper. But still, it is very surprising that the GEOS-Chem model is now found to overestimate HCHO by a factor of 1.47 compared to the same CAMS data. I have serious doubts about the fact that the 24% increase in HCHO yield in the ISOPO2+NO reaction can lead to an overestimate of 47%, and if so, this should be demonstrated.

Before beginning out inversion work, we reproduced the Travis et al. (2016) modeled HCHO along the SEAC4RS flight path (not shown in their publication). The differences between the Zhu et al. (2016) simulation and Travis et al. (2016) simulation include updates to anthropogenic emissions, deposition, the isoprene oxidation mechanism, and the inclusion of alpha-pinene and limonene oxidation mechanisms.

The Zhu et al. (2016) paper finds a -3% model bias in boundary layer HCHO concentrations (page 13483 line 4). In our work, using unscaled isoprene emissions unscaled and the isoprene oxidation mechanism of Travis et al. (2016), the modeled v. measured slope is 1.47 and the intercept is -0.5 ppb, giving a normalized mean bias of 24%.

We find that scaling isoprene emissions down 85% and using the Travis et al. (2016) mechanism gives a normalized mean bias of 16%. Scaling isoprene emissions down 85% and using the Zhu et al. (2016) mechanism gives a normalized mean bias of 8%.

The remaining differences between our work and Zhu et al. (2016) could likely be explained by other updates not included in the Zhu et al. (2016) paper, or by the slightly smaller domain used in this work.

We now include a more detailed list of the differences between this work and Zhu et al. (2016).

p.3, l.7 : I suggest to remove 'older chemical mechanisms and' from the sentence. 'Old' is always relative and for example the latest findings in isoprene chemistry (Bates et al. 2016, Teng et al. 2017, etc.) are not considered in the present study.

This point is well taken, and the phrase has been removed.

In Section 2.2, the effect of soil moisture stress on the bottom-up isoprene fluxes is not discussed. Have you accounted for it in your simulations? The Edwards Plateau in Central Texas is often affected by drought. The strong flux decrease derived in this region might be partly explained by the neglect of soil moisture stress in MEGAN. This warrants some discussion.

We have not accounted for soil moisture stress in our simulations. It is possible that this could explain some of the overestimate in our prior isoprene emissions.

The soil moisture correction applies only when the soil moisture falls below a prescribed wilting point (in units of degree of saturation, expressed as the ratio of soil moisture to the porosity of soil). During the SEAC4RS period, GEOS-FP root zone and top soil wetness were consistently over 0.4 in the Edwards Plateau. This is above the ECMWF wilting point value of 0.171 (Müller et al., 2008). However, the MERRA-2 wilting point, which more closely corresponds to the catchment model used to generate the GEOS-FP soil moisture dataset, is approximately 0.6 in the Edwards Plateau. Previous simulations including the soil moisture activity parameter have found

a pronounced difference in Texas, but little changes in other regions of the southeastern US (Sindelarova et al., 2014).

We have the following to our discussion of the Edwards Plateau:

**"Uncertainty in the dependence of isoprene emission on soil moisture could also affect isoprene emission estimates for the Edwards Plateau (Sindelarova et al., 2014)."**

p.6, l.2 : I'm a bit confused here. In Travis (2016), not only mobile NOx emissions are reduced by 60% but all non-power plant sources (or alternatively 30% reduction of non-power plant sources and no soil NO emissions). In Chan Miller et al. (2017) a decrease of 50% in anthropogenic NOx emissions relative to NEI 2011 is applied. Please clarify what you actually did in the present work and why. Note furthermore that soil NO emissions were found to be too low over the Ozarks by Wolfe et al. (2015).

We apologize for the confusion. In our simulation, we reduce all anthropogenic sources of  $NO_X$  other than power plants by 60%. This results in a 50% reduction of total anthropogenic  $NO_X$  relative to NEI. The downscaling of anthropogenic  $NO_X$  is consistent between Travis et al (2016), Chan Miller et al. (2017), and this work. This has been clarified in the text.

We also reduce soil NO emissions by 50% as in a Travis et al. (2016), which is based on the previous work of Vinken et al. (2014). We have included a note in the text which mentions the uncertainty in soil NOx emissions.

p. 6, l. 15 : Still, it would be useful to give percentage estimate of contribution of other NMVOCs to the formaldehyde columns in Southeast US.

In a simulation without isoprene emissions, maximum HCHO columns are near  $1 \ge 10^{16}$  molecules cm-2, which is about  $5 \ge 10^{15}$  cm-2 higher than the background concentrations over the ocean. This enhancement is less than 20% of the total enhancement in HCHO column observed using MEGANv2.1 uniformly scaled by 0.85. This is consistent with previous analysis (Millet et al., 2006; Palmer et al. 2003).

We have changed the text to read:

**"Non-methane VOCs other than isoprene contribute less than 20% to the HCHO column enhancements over the Southeast US (Palmer et al., 2003; Millet et al., 2006) and are not optimized as part of the inversion."**

p.7, l. 14 : I don't see what is the motivation of the first sensitivity inversion (with reduced errors). Please explain.

The motivation for the first sensitivity test is to examine the influence of  $S_0$  on the inversion results. In our base inversion configuration, spectral fitting errors are increased relative to the values provided in the OMI-SAO product. The first inversion configuration uses the reported instrument uncertainty, as would be typical in the absence of a validation study.

p.10, 1.4 : 50%, do you mean factor of 2 or 1.5?

Factor of two. We have clarified this in the text.

Figure 1 : The OMI SAO columns over the SEAC4RS period do not look the same as in Figure 5 of Zhu et al. (2016). What is the difference? Why did you use error-weighted means? Please specify how these means are calculated (relative or absolute).

We show OMI SAO columns computed using the GEOS-Chem air mass factor, which gives some of the difference. The remainder can be attributed to error weighting. We show errorweighted means because the observations are weighted according to error in the optimization. The elements of  $S_0$  (i.e., the instrument uncertainty in the base inversion configuration) are used in the figures.

Figure 2 : The MEGAN base emission factors shown in the upper left panel should be the same as in Zhu et al. (2016) (both use Hu et al. 2015). They are however about 10 times higher that in Figure 3 of Zhu et al. (2016). Please explain.

We thank the reviewer the careful reading. The MEGAN base emissions factors used in this work are the same as in Zhu et al. (2016). The Zhu et al. (2016) colorbar is mislabeled (personal communication).

[revised manuscript text omitted]